# Mitochondrial Introgression, Color Pattern Variation, and Severe Demographic Bottlenecks in Three Species of Malagasy Poison Frogs, Genus *Mantella*

**DOI:** 10.3390/genes10040317

**Published:** 2019-04-23

**Authors:** Angelica Crottini, Pablo Orozco-terWengel, Falitiana C. E. Rabemananjara, J. Susanne Hauswaldt, Miguel Vences

**Affiliations:** 1CIBIO Research Centre in Biodiversity and Genetic Resources, InBIO, Universidade do Porto, Rua Padre Armando Quintas, N° 7, 4485-661 Vairão, Portugal; tiliquait@yahoo.it; 2School of Biosciences, Cardiff University, Sir Martin Evans Building, Museum Avenue, Cardiff CF10 3AX, UK; Orozco-terWengelPA@cardiff.ac.uk; 3Mention Zoologie et Biodiversité Animale, Faculté des Sciences, Université d’Antananarivo, BP 906, Antananarivo 101, Madagascar; frabemnjr@gmail.com; 4Zoological Institute, Technische Universität Braunschweig, Mendelssohnstr. 4, 38106 Braunschweig, Germany; s.hauswaldt@googlemail.com

**Keywords:** hybridization, *Mantella aurantiaca*, *M. crocea*, *M. milotympanum*, *M. madagascariensis*, *M. pulchra*, cytochrome b, *RAG-1*, microsatellites

## Abstract

Madagascar is a biodiversity hotspot particularly rich in amphibian diversity and only a few charismatic Malagasy amphibians have been investigated for their population-level differentiation. The *Mantella madagascariensis* group is composed of two rainforest and three swamp forest species of poison frogs. We first confirm the monophyly of this clade using DNA sequences of three nuclear and four mitochondrial genes, and subsequently investigate the population genetic differentiation and demography of the swamp forest species using one mitochondrial, two nuclear and a set of nine microsatellite markers. Our results confirm the occurrence of two main mitochondrial lineages, one dominated by *Mantella aurantiaca* (a grouping supported also by our microsatellite-based tree) and the other by *Mantella crocea* + *Mantella milotympanum*. These two main lineages probably reflect an older divergence in swamp *Mantella*. Widespread mitochondrial introgression suggests a fairly common occurrence of inter-lineage gene flow. However, nuclear admixture seems to play only a limited role in this group, and the analyses of the *RAG-1* marker points to a predominant incomplete lineage sorting scenario between all five species of the group, which probably diverged relatively recently. Our demographic analyses show a common, severe and recent demographic contraction, inferred to be in temporal coincidence with the massive deforestation events that took place in the past 1000 years. Current data do not allow to conclusively delimit independent evolutionary units in these frogs, and we therefore refrain to suggest any taxonomic changes.

## 1. Introduction

Madagascar is home to an almost unparalleled diversity of flora and fauna, with the majority of its native species being endemic to this island [1]. Amphibians are represented on Madagascar by five frog clades of independent origin [2,3]. At least some of these clades are very rich in species, such as the endemic family Mantellidae and the endemic microhylid subfamily Cophylinae [4,5], but a large portion of this species diversity remains to be taxonomically described [6,7]. While the taxonomic knowledge on Malagasy amphibians has substantially progressed in the last thirty years [2,8,9] and a large research effort in more recent times has been directed to study the processes of species diversification acting on these radiations [4,10,11,12,13], the population-level differentiation of Malagasy amphibians is still poorly explored. Madagascar is home to many endemic and endangered frog species [9] and this approach can provide useful information to improve the management of the many threatened species. So far, this important aspect of conservation genetics has been investigated mainly in a few charismatic species such as the large tomato frogs [14], the rainbow toad [15], and some Malagasy poison frogs of the genus *Mantella* [16,17].

Phylogeographic studies in amphibians often identify limited gene flow among populations and a genetic structure that correlate with their geography and distribution [18,19,20], which usually is interpreted as a signal of low vagility. In Madagascar poison frogs, this pattern has been observed in some species (e.g., *Mantella bernhardi*, *M. expectata* and, to a shallower extent also in *Mantella viridis* [15,16,21]). On the contrary, other studies have identified a surprisingly low degree of phylogeographic structure and common mitochondrial haplotype sharing in *Mantella* species of the eastern rainforest [22,23]; rapid range expansions in other *Mantella* species and in the tomato frogs of the genus *Dyscophus* [14,17], and widespread co-occurrence of deeply divergent mitochondrial haplotype lineages within the same population that seems to characterize an arid adapted *Mantella* species [24,25].

Due to their usually bright aposematic coloration (a case of convergent evolution with South American poison dart frogs [26,27,28], diurnal activities, and high prevalence in the pet-trade [29,30] *Mantella* are probably among the best-studied group of Malagasy amphibians [17,21,22,23,24,31,32,33]. Yet, information on their distribution and habitat preferences is still lacking for many species [34,35,36,37], and several studies suggest the genus needs a taxonomic revision [16,19,21,24,25].

Within the genus *Mantella* it is striking that often, sympatric species share very similar color pattern, which might indicate phenomena of Müllerian mimicry [38]. This is most obvious in the species pair *M. baroni* and *M. madagascariensis* which belong to different species groups; but also the syntopic *M. laevigata* and *M. manery* share a dorsal pattern similar to each other, as do the uniformly orange *M. aurantiaca* and *M. milotympanum* which live in areas only few km apart. On the other hand, species that apparently are very closely related to each other can strongly differ in coloration [17,21,22] pointing to a probably fast and repeated evolution of chromatic characteristics [22]. Most of the studies to date were entirely based on mitochondrial genes or single nuclear markers which might be affected by introgression phenomena that could blur true relationships; in fact, hybridization has already been confirmed between *M. cowani* and *M. baroni* [17], and recent gene flow was detected between the morphologically distinct *M. nigricans*, *M. baroni* and *M.* aff. *baroni* [17].

In this study we focus on the *Mantella madagascariensis* group that is characterized by a particularly high color variation, both among species and among supposedly conspecific populations. We deem this characteristic favoured the description of different color morphs as distinct species (e.g., in the case of the poorly differentiated *M. crocea* Pintak and Böhme, 1990 and *M. milotympanum* Staniszewski, 1996) and we investigate the population genetics of this group to explore this hypothesis. According to current taxonomy, this group consists of two rainforest species, *M. madagascariensis* and *M. pulchra,* and three swamp forest species *M. aurantiaca, M. crocea* and *M. milotympanum* [31]. *Mantella aurantiaca* is characterized by a uniform and translucent orange coloration (Appendix A). *Mantella milotympanum* from the type locality near Fierenana is opaque red orange with a black tympanic region (the characteristic that is also giving the name to this species) and a ventral surface uniformly orange. *Mantella crocea* shows a lateral black coloration, a frenal stripe, a ventral black color with a variable number of markings that can be grey, bluish-white or yellowish, and a dorsal color being yellow-brown in the more southern populations and bright green in the populations of the western and northern portion of its range. Populations with intermediate patterns between *Mantella crocea* and *M. milotympanum* (Ambatombolana, Andriambe, Savakoanina and Andaingo [35]) were assigned to *M. milotympanum* (Ambatombolana, Andriambe, Savakoanina) based on the presence of the black tympanic region and a uniformly orange ventral surface and on the lack of both lateral black coloration and frenal stripe. The population of Andaingo was assigned to *M. crocea* based on the presence of a lateral black coloration, a frenal stripe and a ventral black coloration with sparse grey, yellowish-white markings (see Appendix A for all details). Despite some overlap in habitat specialization of rainforest vs. swamp forest species, the latter are most common in a stretch of the northern central east of Madagascar characterized by extended swamps surrounded by forests, often with many *Pandanus* screw palms (Figure 1). These species are characterized by distinct alkaloid profiles [39], probably due to low prey electivity and specific arthropod sources in their habitat [40]. As far as known, species in this group usually do not occur syntopically, but their ranges have various contact zones. These frogs are known to hybridize in captivity and haplotype sharing has been reported among *Mantella crocea* and *M. aurantiaca,* and among *M. crocea* and *M. milotympanum* [22,23]. *Mantella milotympanum* and *M. aurantiaca* are currently categorized as Critically Endangered following the red-listing criteria of the International Union for Conservation of Nature (IUCN) [41]. This is mostly due to their restricted and severely fragmented distribution (with an area of occupancy of less than 10 km^2^), the ongoing decline in the extent and quality of their habitat and their possible over-exploitation in the pet-trade. On the contrary, *M. crocea* is listed as of Least Concern due to its relatively wide extent of occurrence and presumed large population [41]. Due to the widespread mitochondrial haplotype sharing between *M. crocea* and *M. milotympanum* (and to a lesser extent between *M. crocea* and *M. aurantiaca*) and their radically different IUCN Red List categorization it is important to investigate the degree of differentiation of these species and assess their taxonomic status, given that most studies to date focused on only mitochondrial genes and did not cover the entire known distributional range of these species.

Although there is still no consensus on the time of human colonization of Madagascar ([42,43] and literature cited therein for more details), a recent study finds no support for human activity before 1500 y B.P., and otherwise finds diverse evidence of human activity from about 1350 y B.P. [43]. It is expected that deforestation increased sharply after human colonization and that this markedly affected the demographic history of Madagascan native species. To test this hypothesis, we investigated the demographic history of the populations of the three swamp forest species of the *M. madagascariensis* group for which we had a complete extended molecular dataset.

To summarize, in this study we investigate the genetic differentiation and demography of species in the *M. madagascariensis* group, in particular targeting the three swamp forest species. We first ascertain monophyly of this this focal group of species by a genus-wide analysis of multiple nuclear and mitochondrial DNA sequences, and subsequently analyse population genetic differentiation in several species of the group based on an extended data set with samples from new localities, and with the inclusion of new molecular markers, *RAG-1*, *RAG-2* and a set of nine microsatellites [44].

## 2. Materials and Methods

### 2.1. Tissue Sampling, DNA Extraction, and Amplification

Two-hundred and fifteen tissue samples of individuals of the *M. madagascariensis* group (sensu [38]) were analyzed in this study, including: 68 samples from four localities of *M. aurantiaca*, 44 samples from four localities of *M. milotympanum*, 72 samples from eight localities of *M. crocea*, 22 samples from three localities of *M. madagascariensis*, and 9 samples from one locality of *M. pulchra* (Figure 1; Appendix A for more details). These covered a large proportion of the known distribution of *M. aurantiaca*, *M. milotympanum* and *M. crocea* [2,35], with the northernmost population of *M. crocea* in the Makira Reserve (locality Hevirina) extending the known distribution of this species substantially. Twenty-three samples of *Mantella* spp. representing all the currently described species and the recently identified cryptic lineages were used for the phylogenetic analyses (details below).

Tissue samples were either toe clips from subsequently released individuals, saliva swabs (*M. crocea* from Zahamena) or femur muscle from collected voucher specimens, all collected between 2002 and 2009 during several field expeditions along the eastern coast of Madagascar and stored in 96% EtOH. Sampling and tissue sample export permits were approved by the Ministère de l’Environnement et des Forêts of Madagascar.

Total genomic DNA was extracted using a proteinase K (10 mg/µL) digestion following a standard saline method [45]. For the phylogeographic analyses (dataset available in Appendix A), polymerase chain reactions (PCR) were used to amplify a fragment of the mitochondrial cytochrome b (*COB*) gene of ca. 600 nucleotides, a fragment of the nuclear recombination activating gene 1 (*RAG-1*) of ca. 750 nucleotides, and a fragment of the nuclear recombination activating gene 2 (*RAG-2*) of ca. 500 nucleotides (Appendix A). In addition to *COB*, *RAG-1* and *RAG-2*, a fragment of the brain-derived neurotrophic factor (*BDNF*), and of the mitochondrial cytochrome oxidase subunit 1 (*COX1*), 12S rRNA and two fragments of 16S rRNA genes were sequenced for all known *Mantella* species to ascertain the monophyly of the *M. madagascariensis* group within the genus (for primer sequences, see Appendix A).

PCRs were performed in a final volume of 11 µL and using 0.3 µL each of 10 pmol primer, 0.25 µL of total dNTP 10 mM (Promega, Madison, WI, USA), 0.08 µL of 5 U/mL GoTaq^®^, and 2.5 µL 5x Green GoTaq^®^ Reaction Buffer (Promega). Primers and PCR amplification conditions in Appendix A. The successfully amplified double-stranded PCR products, treated with Exonuclease I (New England Biolabs, Ipswich MA, USA) and Shrimp Alkaline Phosphatase (Promega) to inactivate remaining primers and dNTPs, were directly used for the cycle-sequencing reaction using dye-labeled dideoxy terminators (Applied Biosystems, Foster City, CA, USA) with the amplification primers on an ABI 3130xl automated DNA sequencer. All nuclear genes were sequenced in both directions.

The same tissue samples that were used for the phylogeographic analyses were genotyped for nine microsatellite loci (Appendix A). Details on the primer combinations used, the amplification, and the genotyping protocols are described in [44]. For a complete list of species, collecting localities, coordinates, gene fragments and microsatellite used see Appendix A.

### 2.2. Sequence Alignment, Statistics, and Phasing

Chromatographs were checked and sequences were edited and aligned using Codon-Code Aligner (v. 2.0.6, Codon Code Corporation, Centerville, MA, USA). A homologous *COB* sequence of *M. baroni* (AY862201) was added to the *COB* alignment. A Neighbor-Joining (NJ) tree based on uncorrected p-distances was constructed with MEGA, version 7.0.9 [46] to gain a first overview of differentiation among sequences (Appendix A). We also added to this alignment 17 unpublished shorter *COB* sequences of *M. aurantiaca* from the captive-bred colony housed in the breeding facility managed by Association Mitsinjo in Andasibe) (Appendix A). All newly obtained sequences ware deposited in GenBank (MK422202-MK422436; MK440330-MK440540).

We discarded the hypothesis of pseudogenes occurring in the *COB* sequence dataset by confirming the absence of stop codons in the analyzed fragment. Because the *RAG-1* and *RAG-2* alignments contained heterozygous positions we phased genotypes to identify haplotypes for further analyses using the PHASE algorithm (version 2.1.1) with default settings [47] as implemented in the software DnaSP (version 6.12.01; [48]). PHASE parameters were 1000 iterations, one thinning interval and 100 burn-in iterations and a posterior probability threshold of 0.9 to determine the most probable inferred haplotypes for each nuclear sequence. Analyses were repeated three times using different seed values. We determined the number of base substitutions using DnaSP for the *COB*, *RAG-1* and *RAG-2* fragments and computed Tajima’s average number of pairwise differences (∏; [49]) and haplotype diversity (Hd; [50]) for each locality and for all fragments used for the phylogeographic analyses (Appendix A). We also computed Tajima’D [51], computed as the difference between the mean number of pairwise differences and the number of segregating sites, for the populations that were used to investigate the demographic history of the three swamp forest species (details below).

### 2.3. Phylogenetic Analysis

DNA sequences of four mitochondrial and three nuclear genes were sequenced from one individual (voucher number available in Figure 2) of each *Mantella* species, to verify monophyly of the *M. madagascariensis* group, which contains the focal taxa of this study. Sequences were aligned as described above (nuclear sequences not phased) and concatenated for analysis. Because mitochondrial introgression has previously been detected in *Mantella* [22] we analyzed nuclear and mitochondrial genes separately. The alignment of the three nuclear genes was analyzed as a single partition to avoid overparametrization, under a TIM1+I substitution model selected by the AIC criterion in jModeltest [52]. For the concatenated mitochondrial genes, a partition scheme with seven partitions was selected with PartitionFinder [53], as follows: (1) 1st positions of *COB* and 3rd position of *COX1*, HKY+I; (2) 2nd position of *COB*, GTR+G; (3) 12S and 3rd position of *COB*, GTR+I+G; (4) 1st position of *COX1*, GTR+G; (5) 2nd position of *COX1*, SYM+I; (6) 16S, GTR+I; (7) intervening tRNAs (tRNA Leut, tRNA Val) between 16S and 12S, GTR+G. Bayesian inference (BI) phylogenetic analysis was subsequently run in MrBayes [54], with 50 million generations and a burn-in of 25% (Figure 2).

### 2.4. Haplotype Network Reconstruction

Individual *COB* sequences and phased allelic sequences for the *RAG-1* and *RAG-2* gene fragments were collapsed into haplotypes using the online web tool DnaCollapser 1.0 available on the FaBox platform (http://users-birc.au.dk/biopv/php/fabox/). Prior to calculating haplotype relationships using networks, we analyzed the nuclear *RAG-1* and *RAG-2* datasets with the Phi test of Bruen et al. [55] as implemented in SplitsTree4 [56] to assess the presence of recombination in the data. Neither of the genes presented significant evidence of recombination (*p*-values > 0.1) and thus they were used for network reconstruction. We used TCS version 1.21 [57] to analyse the relationships of the *COB* haplotypes and of the *RAG-1* and *RAG-2* inferred haplotypes (Figure 3, Figure 4 and Figure 5). With these networks we show instances of haplotype sharing and highlight haplotype differences between and within analyzed populations, providing an intuitive graphic representation of the observed differentiation between populations, but being aware that phased haplotypes might not in all cases be correct due to the probabilistic nature of the Phase algorithm, and due to the large haplotype variation, the reconstructed haplotype relationships might not be fully accurate. The method underlying TCS [58] calculates the number of mutational steps by which pairwise haplotypes differ, computing the probability of parsimony for pairwise differences until the probability exceeds 0.95. The manual adjustment of the threshold was necessary to infer the number of mutational steps that were needed to join haplotypes into one single network for the analyzed *COB* fragment. Loops that occurred in the network were resolved by (1) treating connections with singletons and/or rare haplotype to be less likely than connections with central and frequent haplotypes [59] and (2) using the geographical criterion, favouring connection between haplotypes of the same or close population over connection with haplotypes from distant populations [60].

### 2.5. Microsatellite Analyses

The presence of null alleles was tested with MICRO-CHECKER [61]. Linkage disequilibrium between loci was tested using ARLEQUIN 3.5.1.2 [62] with 10^4^ permutations to assess significance. The average number of alleles per locus, allelic richness (AR), expected (H_E_) and observed (H_O_) heterozygosities and inbreeding coefficients (F_IS_) were calculated using MSA v. 4.05 [63] (Table 1). Allelic richness was calculated based on a minimum of 11 individuals per population. GENEPOP [64] was used to assess the Hardy–Weinberg equilibrium of each locus and population using 1000 dememorizations, 100 batches and 1000 iterations per batch.

### 2.6. Population Structure

The mitochondrial DNA based analyses reflect only the maternal genealogy relating these *Mantella* populations; therefore, we carried out an analysis of population structure using the nine nuclear microsatellites to attempt to identify the best way of grouping the individuals with a dataset with higher statistical power and representative of both parental lineages. This analysis was carried out with the software Bayesian Analysis of Population Structure (BAPS) [65] (Figure 6). Only individuals with ~10% or fewer missing data were kept for this analysis to reduce potential noise caused by the missing data (e.g., spurious clusters). This reduced the microsatellite dataset to 129 samples (Table 1). A range of possible values (1 to 10) for the optimal number of clusters in the data (K) was tested using the group-based approach in BAPS, and a-priori grouping individuals according to their sampling localities. The analysis was repeated three independent times to assess repeatability of the results. A NJ tree describing the similarity between sampling localities was built using Phylip [66] with 100 bootstrap replicas to assess branching pattern statistical support on the basis of the genetic distance (Figure 7). Proportion of Shared Alleles [67] was calculated in MSA. Pairwise F_ST_ between sampling localities was also computed with MSA with 10^4^ permutations to assess significance (Table 2). We also used BayesAss v. 3.0.4 [68] to estimate migration patterns between populations. BayesAss uses an MCMC algorithm to estimate the proportion of individuals form a population that derives from a different source population. This is achieved by estimating the likelihood of ancestry for each individual from each population analyzed using assignment methods that use the information on the individual’s genotype and the allele frequency distributions in the different populations. BayesAss’ MCMC running parameters were burn-in for 25 million steps, followed by 25 million steps of data collection saving each 1000th step. BayesAss was run in triplicate to assess repeatability of the results inferred with the MCMC algorithm. Convergence of the Markov chains was determined using Tracer from the BEAST suite [69].

### 2.7. Demographic History

Due to the conservation concerns on these species we carried out analyses of the demographic history of these populations to determine whether they present robust effective populations sizes that have been stable across time and thus maintained their evolutionary potential. For this purpose, the demographic history of the nine localities with the largest sample sizes and no more than 10% missing data was estimated with Msvar 1.3 [70,71]. Msvar uses the counts of alleles for each locus in a population to estimate whether the current effective population size (N0) is explained by a demographic change (t) generations ago from an ancestral effective population size (Nt), e.g., an expansion or a bottleneck. Msvar uses coalescent simulations conditioned on prior distributions for each of the demographic parameters, as well as the mutation rate of the microsatellites. For these analyses we carried out runs under different prior combinations representing bottlenecks, expansions and stable demography (no change in effective population size) for each population to determine that the posterior estimates of the parameters were not dependent on the underlying demographic model. The prior and hyperprior distributions used are shown in Appendix A. A generation length of one year was assumed for all species and an average vertebrate microsatellite mutation rate of 10^−4^ [72,73,74] allowing it to vary between 10^−2^ and 10^−6^. Each Msvar was run for each demographic scenario for 400 million steps of the MCMC algorithm. For each sampling locality three independent runs (one under each demographic prior) were analyzed for convergence using Gelman and Rubin’s statistic [75] implemented in the library CODA [76] in R [77], after discarding the initial 50% of the MCMC steps as burn-in phase.

## 3. Results

### 3.1. Monophyly of the M. madagascariensis Group

To understand if the target species of this study, defined as the *M. madagascariensis* group (*M. aurantiaca, M. crocea, M. madagascariensis, M. milotympanum, M. pulchra*) together form a monophyletic group within the genus *Mantella*, we performed a phylogenetic analysis based on DNA sequences of multiple genes. To exclude effects of mitochondrial introgression, we analyzed mitochondrial and nuclear genes separately. The phylogenetic trees obtained from BI analysis of the concatenated nuclear and of the mitochondrial data sets (Figure 2a,b) both resolved the deep phylogenetic relationships in the genus concordantly. *Mantella bernhardi* was placed as sister group to the remaining species of *Mantella*; of these, the *M. cowani* group (*M. baroni*, *M. cowani*, *M. haraldmeieri*, *M. nigricans*) was sister to a clade containing the *M. betsileo* group (*M. betsileo*, *M. ebenaui*, *M. expectata*, *M. viridis*), *M. laevigata* group (*M. laevigata*, *M. manery*) and *M. madagascariensis* group. The nuclear DNA analysis failed to unambiguously resolve relationships among the *M. laevigata* and *M. betsileo* groups, probably due to missing data for *M. manery* for which only a *RAG-1* sequence was available. Similarly, relationships within the species groups did not receive significant support in the nucDNA analysis. However, both analyses provided high support for monophyly of the *M. cowani* group and of the *M. madagascariensis* group.

### 3.2. DNA Sequence Diversity and Differentiation within the M. madagascariensis Group

The total alignment length (after trimming the ends of the alignment to remove missing data) for the three genes was: *COB*, 215 individual sequences and a total length of 552 nucleotides; *RAG-1*, 211 individual sequences (422 after phasing) and a total length of 732 nucleotides; *RAG-2*, 21 individual sequences (42 after phasing) and a total length of 531 nucleotides. In all the three alignments we did not detect any gaps, and the translation into amino acids resulted in neither nonsense nor stop codons. In the *COB* alignment we found 81 (14.7%) variable sites (out of 552), among which 69 were parsimony-informative and 12 were singletons, and a total of 49 haplotypes were identified (C01-C49; see Appendix A for more details). In the phased *RAG-1* alignment 59 sites (out of 732) were variable (8.1%), among which 51 were parsimony-informative and 8 were singletons, and a total of 78 haplotypes were identified (R01–R78; Appendix A). In the phased *RAG-2* alignment 27 sites (out of 531) were variable (5.1%), among which 16 were parsimony-informative and 11 were singletons, and a total of 17 haplotypes were identified (01–17; Appendix A). In the analyzed *COB* fragment the Hd values of the populations of *M. crocea* were high (ranging from 0.6 and 1 in *M. crocea*), while the Hd values of the analyzed populations of *M. aurantiaca* and *M. milotympanum* were very variable, ranging from 0 to 0.684 in *M. milotympanum* and from 0 to 0.752 in *M. aurantiaca* (Appendix A). The nucleotide diversity (π) values for the localities provide an estimate on the genetic variability in a given data set. In the *COB* alignments the *M. aurantiaca* populations from Andranomandry and Besariaka show the highest values of π (see Appendix A). In the *RAG-1* and *RAG-2* fragment estimated Hd values were on average high, ranging from 0.533 and 1 in all analyzed populations (Appendix A). *RAG-1* nucleotide diversity (π) values were on average lower than the estimated π at the *COB* fragment for *M. aurantiaca* and *M. crocea* and higher for *M. milotympanum*, *M. madagascariensis* and *M. pulchra* (see Appendix A). Although the number of analyzed sequences per locality differed across the different localities (ranging from 2 to 52 in the phased *RAG-1* alignment), it is worth noting that in this alignment the highest π was observed in the population from Ampangadimbolana of *M. crocea* (see Appendix A). Nucleotide and haplotype diversity values for each locality per each analyzed locus are provided in Appendix A. The values of Tajima’s D computed for the *COB* fragment are mostly negative (see Appendix A), suggesting population size expansion although this result is likely to reflect the occurrence of widespread mitochondrial introgression between the analyzed populations. On the other side, the values of Tajima’s D computed for the *RAG-1* and *RAG-2* fragments are mostly positive (see Appendix A), and are in overall agreement with the results obtained in the demographic analyses performed with the microsatellite data (see details in Section 3.6), suggesting a decrease in population size.

The NJ tree (Appendix A) based on the *COB* sequences suggests several instances of introgression of the *M. crocea* mitochondrial genome into the populations of *M. aurantiaca* of Andranomandry and Besariaka, and of introgression of *M. pulchra* mitochondrial genome into the population of *M. aurantiaca* of Torotorofotsy (see also Appendix A). A probable introgression of an *M. aurantiaca* mitochondrial genome was identified for the population of *M. crocea* from Ambohitantely, but the respective sequence differed by four mutations from the nearest haplotype identified in the extant populations of *M. aurantiaca* sampled by us.

The *COB* sequences of *M. milotympanum* and *M. crocea* were poorly differentiated from each other, and in this group the most differentiated clade was composed by sequences originated from the recently identified population of *M. crocea* in Hevirina (in the Makira area), the northernmost population known for this species. This analysis also shows that the northern populations of *M. madagascariensis* share the mitochondrial genome with *M. pulchra* (Appendix A). The 17 shorter sequences of *M. aurantiaca* coming from the captive bred colony showed almost no genetic differentiation to the population of *M. aurantiaca* from Totorotofotsy (Appendix A). As already observed in some individuals of *M. aurantiaca* from Andranomandry and Besariaka, also one individual of this pool carried the mitochondrial genome of *M. crocea* (Appendix A), suggesting that introgression has also taken place in these northern populations of *M. aurantiaca*, and not only in the southern ones.

### 3.3. Haplotype Networks

The haplotype network of the *COB* alignment led to the identification of three unconnected subnetworks plus one additional disconnected haplotype (C41) (Figure 3). One network was mostly composed of sequences of *M. madagascariensis* and *M. pulchra,* a second network contains all the sequences of *M. milotympanum*, most of the sequences of *M. crocea* (both from the brown and green phenotype populations) and some sequences of *M. aurantiaca* from Andranomandry and Besariaka. The third network is mostly composed of sequences on *M. aurantiaca* from its entire distributional range, but it also includes one haplotype of *M. crocea* from Ambohitantely (green phenotype; C17; see Appendix A). Haplotype C41 requires at least 11 mutational steps to be linked to the *M. aurantiaca* network (haplotype C10), and at least 23 mutational steps to be linked to the *M. crocea*/*M. milotympanum* network (haplotype C13; Figure 3). Haplotype C41 is shared among all the analyzed individuals of *M. madagascariensis* from Ranomafana suggesting that this population is quite differentiated from the other analyzed populations, although the limited number of included populations and specimens of this population limits our conclusions. Twenty-three mutational steps are needed to link the network of the *M. pulchra*/*M. madagascriensis* to the network of the *M. crocea*/*M. milotympanum.* In the *M. crocea*/*M. milotympanum* network, the individuals of the population from Hevirina (that share the green phenotype with specimens from Ambohitantely, Zahamena and Andaingo) group together, and 10 mutational steps link this group of haplotypes to all the others. See Appendix A and Figure 1 and Figure 3 for more details on haplotype frequencies and distribution.

Only one network was retrieved for the phased *RAG-1* alignment, which included all the five species of the *M. madagascariensis* group (Figure 4). The three individuals of *M. aurantiaca* from Andranomandry that show sign of mitochondrial introgression with *M. crocea* (see Figure 3, Appendix A) share two of the most widespread *RAG-1* haplotypes (R1, shared by all species of the *M. madagascariensis* group, and R3, shared by all species of the *M. madagascariensis* group with the exception of the southern population of *M. madagascariensis*, see Appendix A and Figure 4), but also haplotype R6, which is exclusive of *M. aurantiaca* populations of Andranomandry and Besariaka. In the population of *M. aurantiaca* from Besarika, nine of the 21 analyzed individuals show mitochondrial introgression with *M. crocea* (see Figure 3, Appendix A). Most of these introgressed individuals, for *RAG-1*, bear haplotype R2, which is shared by individuals of all the analyzed localities of *M. aurantiaca*, but also by *M. milotympanum*, *M. crocea* (green phenotype populations) and *M. madagasariensis*. Haplotype R10 and R15 are uniquely found in these individuals. Other *RAG-1* haplotypes found in these introgressed individuals are: R1, widely shared by all analyzed species; R3, shared by all species of the *M. madagascariensis* group with the exception of the southern population of *M. madagascariensis*; R4, mostly found in the brown populations of *M. crocea* but also found in *M. milotympanum*, *M. aurantiaca* and in the southern population of *M. madagascariensis*; R5, mostly found in *M. aurantiaca* populations from Andranomandry and Besariaka, but also found in the *M. crocea* population from Andasibe and in the population of *M. madagascariensis* from Ranomafana; R6, exclusive of *M. aurantiaca* populations of Andranomandry and Besariaka; R12, mostly found in *M. milotympanum* but also found in the *M. aurantiaca* population of Besariaka, in the population of *M. crocea* of Ambohitantely (green phenotype) and in the southern population of *M. madagascariensis* (Figure 4). The individual of *M. aurantiaca* from Torotorofotsy which shows signs of mitochondrial introgression from *M. pulchra,* shares *RAG-1* haplotype R30 with another *M. aurantiaca* individual from Torotorofotsy, with *M. crocea* from Ambodivoasary and Andasibe (brown phenotype) and with *M. pulchra* (Figure 4). In the *RAG-1* network, the individual of *M. crocea* from Ambohitantely with introgressed *M. aurantiaca* mitochondrial genome shares haplotype R1 with all the other species of the *M. madagascariensis* group, and R28 mostly with individuals of *M. crocea* from Ambohitantely (green phenotype) and Ampangadimbolana (brown phenotype), but also with another individual of *M. aurantiaca* from Torotorofotsy (Figure 4).

The specimens of *M. madagascariensis* from the North (Besariaka) mostly share haplotype R18 (found also in one population of *M. aurantiaca* and *M. milotympanum*) and R19 (found also in one population of *M. aurantiaca* and in the population of *M. madagascariensis* of unkown origin). Other *RAG-1* haplotypes found in this population are: R21, also found in one population of *M. aurantiaca* and *M. milotympanum* and in the population of *M. madagascariensis* of unkown origin); R67, also found in the population of *M. madagascariensis* of unkown origin; and R68, uniquely found here (Figure 4). In the population of *M. madagascariensis* of unknown origin (but showing a pattern of genetic variability similar to the population of *M. madagascariensis* from Besariaka) we found the following *RAG-1* haplotypes: R1, R2, R3, R19, R21 and R67 as discussed above; R65, found also in the population of Andriambe of *M. milotympanum* and *M. pulchra*; and R73, uniquely found in this population (Figure 4).

All the *RAG-1* haplotypes group together in a single network, but there are a few groups of haplotypes that are both uniquely found in one population and are separated by at least 3 mutational steps by all the rest: (*i*) haplotypes R36, R40, and R41 are uniquely found in the population of *M. crocea* from Ambohitantely; and (*ii*) R57 and R56 in the *M. crocea* population of Hevirina (both of them showing the green phenotype); (*iii*) haplotype R47 was found in the population of *M. crocea* from Ampangadimbolana (with the brown phenotype); (*iv*) haplotypes R10, R13 and R17 plus haplotypes R49 and R50, and haplotype R59, are found in the population of *M. aurantiaca* from Besariaka, *M. crocea* from Ampangadimbolana (brown phenotype) and *M. crocea* from Zahamena (green phenotype), respectively. Similar to what was observed in the *COB* network, the population of *M. crocea* showing a more consistent genetic differentiation from the others is the northernmost known population of Hevirina (cf. Figure 3 and Figure 4; *RAG-1* haplotypes R56 and R57). See Appendix A and Figure 4 for more details on *RAG-1* haplotype frequencies and distribution.

The *RAG-2* fragment was amplified only for a small subset of individuals from six populations (some samples of *M. aurantiaca* from Andranomandry, of *M. crocea* from Ampangadimbolana, of *M. milotympanum* from Fierenana-Sahamarolambo, Andriambe and Savakoanina, and of *M. madagascariensis* from Besariaka), mostly to verify if other nuclear genes display a pattern of differentiation comaparable to that in *RAG-1*. The phased *RAG-2* sequences grouped into a single network, although the genetic differentiation of the *RAG-2* haplotypes might be slightly higher compared to *RAG-1* (Figure 5). See Appendix A and Figure 5 for more details on *RAG-2* haplotype frequencies and distribution.

### 3.4. Population Genetic Variation in Microsatellites

We tested for Hardy–Weinberg Equilibrium (HWE) in the nine microsatellite markers studied by sampling locality. At all localities included in the microsatellite analysis, except for the *M. crocea* population from Ambodivoasary, the data revealed absence of HWE (Fisher Exact test *p*-value < 0.001). In all cases this lack of HWE was due to a deficit of heterozygotes, in particular in loci G10X, B8 and M14. Such pattern could be indicative of the presence of null alleles on these loci, however, we found no significant difference in estimates of genetic variation and inbreeding coefficients when excluding them (Welch corrected *t*-tests *p*-value > 0.1), thus we retained these loci for further analyses. No significant linkage disequilibrium within any of the sampling localities was detected after Bonferroni correction for multiple testing, with the exception of the pairwise comparison between loci A9 and B7 in the *M. milotympanum* population of Andriambe (*p*-value < 0.0001). Excluding the *M. crocea* sampling site of Ambodivoasary due to its small sample size, the number of alleles per locus ranged between a minimum of 6.11 and 12.78 across localities, the observed and expected heterozygosity were relatively high, ranging between 0.46 and 0.74, and 0.77, and 0.88, respectively (Table 1). The inbreeding coefficient ranged between a minimum of 0.14 and a maximum of 0.45 (Table 1) in the different localities.

### 3.5. Population Structure

BAPS identified seven clusters as the most probable clustering solution in the data using prior information on the sampling locality of the individuals (PP, posterior probability of each of the analysis replicates > 0.99). The same results were obtained for three independent BAPS runs indicating strong support for the pattern of population structure identified. These included three clusters of *M. aurantiaca*, i.e., (1) Andranomandry and Besariaka, (2) Sahasarotra, and (3) Torotorofotsy. The *M. crocea* samples formed three additional clusters: (4) Ambodivoasary, (5) Ambohitantely, and (6) Ampangadimbolana. Lastly, cluster (7) was formed by the *M. milotympanun* samples from Fierenana-Sahamarolambo, Andriambe and Savakoanina (Figure 6). With the exception of one individual of *M. aurantiaca* from Besariaka, no other sample presented significant admixture (Figure 6). The genetic variation of this admixed individual resembles to 50% cluster 1, 32% cluster 6, 11% cluster 7, 4% cluster 2, 2% cluster 4, and 1% cluster 3 (Figure 6), however, this observation is likely to be an outlier genotype relative to the allelic frequencies expected under Hardy–Weinberg Equilibrium for these *Mantella* populations, rather than the outcome of multiple past hybridisations. These observations are supported by the partition results for progressively larger values of K (i.e., from 2 until 7; Appendix A), which show that from an earlier level on (i.e., K = 2) the localities form well defined clusters, and single individuals appear admixed for the different values of K: the same individual from Besariaka found admixed in the K = 7 solution (Figure 6) was also found admixed in K = 5 and K = 6 solutions (Appendix A); however, different individuals were identified as possibly admixed in the K = 4 and K = 3 solutions. An individual-based clustering approach (without prior locality information) in BAPS rendered incongruent results as no stable clustering solution was identified, further reflecting the effect of the little divergence observed between these localities (e.g., Table 2). Consistent with this hypothesis, a correspondence factor analysis using Genetix [78] of either the individuals, or groups of individuals by localities, resulted in a large overlap between samples and localities, respectively (Appendix A).

Pairwise divergence between sampling localities as measured by the F_ST_ showed that most of the comparisons were significant after Bonferroni correction (Table 2). The F_ST_ values were overall low both within and between species, with the average within species F_ST_ (0.05) slightly smaller than the average between species F_ST_ (0.07). The observed low differentiation between sampling localities was reflected by the average low frequency of private alleles, namely 0.0819. These values reflect either recent gene flow between populations, or rather little divergence from the ancestral population, also reflected by the low statistical support in the dendrogram depicting the relationships between sampling localities (Figure 7). Migration patterns estimated by BayesAss revealed that the majority of the individuals from one locality can be assigned to its sampling locality (between ca. 65–90%), while the remaining individuals correspond to putative migrants from the other localities, each contributing at most up to 5% of the individuals.

### 3.6. Demographic History

For each of the populations included in the microsatellite analysis we used Msvar to estimate its past demographic history using as priors a bottleneck, a population expansion and stable demography (Appendix A). The three independent runs of Msvar were assessed for convergence using Gelman and Rubin’s statistic that for each parameter of the analysis presented values smaller than 1.1 indicating convergence of the demographic model for each locality independently of the prior used. Overall all populations showed evidence of a drastic demographic bottleneck that reduced effective population sizes in some cases from around 100,000 individuals to sometimes a couple of hundreds (e.g., the population of *M. aurantiaca* in Andranomandry; Figure 8, Appendix A). This bottleneck was estimated in very recent times, with the closest estimate to the present at about 87 years ago and the oldest approximately 600 years ago, and an average estimate of approximately 250 years ago (Figure 8).

## 4. Discussion

### 4.1. M. aurantiaca, M. crocea and M. milotympanum—Separate Species or Color Morphotypes?

Among the swamp frog species of the *M. madagascariensis* group, *M. aurantiaca* is the taxon with the most stable coloration, being characterized by a bright orange color on a translucent base (see Figure 1, Figure 6, Figure 7 and Figure 8). Based on their overall coloration, *M. milotympanum* and *M. aurantiaca* were previously thought to be closely related to each other, until genetic data confirmed *M. milotympanum* to be more closely related to *M. crocea* from a mitochondrial perspective [22,23]. Based on these results it has been hypothesized that the orange color might have evolved convergently in *M. aurantiaca* and *M. milotympanum* [22]. Here we confirm that there are two main mitochondrial lineages of which one is prevalent in *M. aurantiaca* and the other in *M. crocea* + *M. milotympanum*, and we detected widespread mitochondrial haplotype sharing between the southern populations of *M. aurantiaca* and the *M. crocea*/*M. milotympanum* group, in *COB* (with two haplotypes, C9 and C5, being shared by *M. aurantiaca* and both brown and green populations of *M. crocea*), *RAG-1* (with widespread haplotype sharing between these taxa) and *RAG-2* (with 1 widespread shared haplotype between *M. aurantiaca* and *M. milotympanum*) (Figure 3, Figure 4 and Figure 5).

In addition to this widespread haplotype sharing in the analyzed mitochondrial marker, it is worth noting the existence of at least two other cases of mitochondrial introgression, with some *COB* haplotypes uniquely identified in the southern populations of *M. aurantiaca* that cluster in the *M. crocea*/*M. milotypmpanum* phylogroup (haplotypes C3, C4 and C8), and one haplotype uniquely identified in the Torotorofotsy population of *M. aurantiaca* that clusters with the *M. pulchra*/*M. madagascariensis* (haplotype C11) network (Figure 3). For the first time, we also identify an instance of mitochondrial introgression of *M. aurantiaca* in *M. crocea* (green population from Ambohitantely), with haplotype C17 being nested within the *M. aurantiaca* network, and therefore confirming mitochondrial introgression happening in both directions between these taxa (Figure 3). *Mantella aurantiaca* was until recently known from 21 localities [35] and its distribution has been currently updated to include 139 breeding sites [37]. The four analyzed populations of *M. aurantiaca* come from both its southern (Andranomandry, Sahasarotra and Besariaka) and northern (Torotorofotsy) edge of the known distribution. Two of the southern populations of *M. aurantiaca* (Andranomandry and Besariaka) formed a single cluster in our microsatellite analysis (Figure 6) suggesting the occurrence of gene flow between these two geographically close populations. Despite this, in our microsatellite-based tree the cluster including the four analyzed populations of *M. aurantiaca* turned out to be the only grouping that received relevant bootstrap support, therefore indicating that the populations of orange translucent color (i.e., *M. aurantiaca*) may form a genetically coherent unit also from a nuclear DNA perspective (Figure 6 and Figure 7).

Different from *M. aurantiaca*, populations of both *Mantella crocea* and *M. milotympanum* are characterized by a great chromatic variability. Individuals of *M. crocea* show a brown-yellowish coloration in the southern and a greenish coloration in the northern part of their range (including the populations of Hevirina, Andaingo, Ambohitantely, and Zahamena analyzed in this study), whereas *M. milotympanum* populations vary greatly in the extent of the black color around the tympanic area and in the overall coloration (predominantly greenish in the southern and orange in the northern part of their range), despite their restricted distributions. Here we confirm fairly common *COB* haplotype sharing between *M. milotympanum* and *M. crocea*. Although the majority of the analyzed individuals of *M. milotympanum* have haplotypes that are poorly differentiated, they are different from the haplotypes found in the populations of *M. crocea* (Figure 3). The analysis of the *RAG-1* marker reveal a deeply interconnected *M. milotympanum*, with only four exclusive haplotypes identified in the analysis of the entire *RAG-1 M. milotympanum* dataset (Figure 4), although probably this marker suggests an overall widespread incomplete lineage sorting between these taxa.

At the interspecific level, our microsatellite-based population structure analyses recover a single *M. milotympanum* population (Figure 6), confirming that the variability of their color morphotypes is not indicative of taxonomic differentiation. In contrast, the three populations of *M. crocea* were recovered as distinct clusters in the BAPS analysis, therefore suggesting that gene flow between color morphs, and more in general between populations, is somewhat limited in this other taxon (Figure 6).

In this study we analyzed the population genetics of three swamp forest poison frogs of Madagascar using a varied panel of molecular markers to infer both recent and more ancient genetic processes of differentiation. We interpret the two main mitochondrial lineages in the three swamp forest species (one prevalent in *M. aurantiaca* the second in *M. crocea/milotympanum*; obvious also from our multi-gene phylogenetic analysis; Figure 2), as reflecting an original, probably allopatric divergence. However, we think that inter-lineage gene flow has been a common phenomenon in the subsequent history of these groups, with probably more ancient mitochondrial introgressions (of *aurantiaca* mtDNA into the Ambohitantely population of *M. crocea*) and recent gene flow (multiple introgressions between neighbouring populations, of *M. crocea* mtDNA into *M. aurantiaca*) (Figure 3). At the same time, we observe that in some cases genetic divergence increases with geographic distance, e.g., in *COB* the haplotypes of the geographically distant, green-colored populations of *M. crocea,* are most divergent (Figure 1, Figure 3 and Figure 4). Despite these evidences, it seems that populations are in part maintaining their genetic identity, with admixture playing only a limited role (Figure 6), and recent migration among populations being rare according to the microsatellite data.

### 4.2. Taxonomic Conclusions

A taxonomic resolution of closely related species complexes such as the *M. madagascariensis* group is inevitably a highly controversial matter. In the studies available to date ([22,23,31,79]; and this study) different kinds of data sets suggested different relationships within this group. Compared to most other species and candidate species of Madagascar frogs [6,7] the genetic distances between valid species of *Mantella* spp. are exceedingly low; and this seems to be the case also in the *M. madagascariensis* group; e.g., in the 16S rRNA, the sequenced specimens of *M. crocea* and *M. milotympanum* have identical haplotypes, *M. aurantiaca* and *M. crocea/milotympanum* differ by 0.8%, and their distance to *M. pulchra* and *M. madagascariensis* is 0.2–1.4%. These values are much lower than the 3% that was set as threshold for initial identification of candidate species for taxonomic revision in neobatrachian frogs [6,80]. Along with the evidence for widespread hybridization between these taxa, this indeed casts doubts on the species status of all these lineages. While we feel that at least some of these deserve being considered as distinct species (e.g., *M. aurantiaca*), the current data do not allow to conclusively delimit independent evolutionary units in the entire complex, and therefore we do not see a simple answer in immediate reach to clarify the uncertain taxonomic status of *M. crocea* and *M. milotympanum*. Given the conflicting genetic results, but also the importance of these species in conservation and their listing in the Convention on the International Trade in Endangered Species (CITES), a certain stability of the (necessarily inconsistent) current taxonomy is important and we therefore here choose not to propose—and strongly discourage—any premature nomenclature and/or taxonomic changes in this group [81].

Independent from their taxonomic status, our data herein suggest that various populations of the *M. madagascariensis* group should be managed as separate, genetically differentiated units for conservation. This refers to the taxonomically named units, but also within *M. aurantiaca* and *M. crocea,* there are genetically divergent clusters of populations that warrant attention in efforts to maintain the genetic variation of these swamp forest frogs.

While the present paper was under final review, Klonoski et al. [82] published a population genomic analysis of *M. aurantiaca*, *M. crocea* and *M. milotympanum* using restriction site-associated DNA (RAD) sequencing. Similar to our results, their study found clear genetic clusters that did not align with current species designations, and supported a substantial genetic differentiation between yellow- and green-colored populations of *M. crocea*. However, the RADseq data [82]—obtained from different individuals and partly different populations than our microsatellite data—found strong indication of admixture at several sites, and an intriguing pattern of genetic clustering of yellow-colored *M. crocea* specimens with *M. aurantiaca*, not recovered by our study. These results reinforce our conclusion that the taxonomy of this complex of frogs is convoluted but without an obvious and simple resolution in sight, despite an increasing body of detailed population genetic and population genomic data.

### 4.3. Mitochondrial Introgression in the Northern Populations of M. madagascariensis?

Samples from the three analyzed populations of *M. madagascariensis* did not form a single mitochondrial network. The southern populations were extremely homogeneous (only one single haplotype was identified from Ranomafana), whereas, the samples of *M. madagascariensis* from the central east (morphologically and chromatically indistinguishable from the population from the South) show haplotype sharing and shallow mitochondrial differentiation with the chromatically different *M. pulchra* (Figure 1 and Figure 3), in apparent agreement with allozyme analyses [31] and karyological data [79]. On the contrary, the haplotype network of the analyzed *RAG-1* fragment show that haplotype sharing between *M. pulchra* and *M. madagascariensis* is extremely rare in this gene and only affects a single haplotype (haplotype R65; Figure 4). The observed pattern may reflect either widespread introgression of divergent haplotypes after speciation, masking the true phylogenetic relationships among taxa and populations [83], or the populations of *M. madagascariensis* from the North can instead be a different color morph of *M. pulchra* (considering that the Type locality of *M. madagascariensis* is “Forêt d’Ambalavatou, entre Mananzarine et Fianarantsoua” it is in fact likely that the population from Ranomafana will be the one maintaining the name *M. madagascariensis*). The Ranomafana population could not be included in the multi-gene phylogenetic analysis (Figure 2) because no more material (tissue, DNA) from these specimens was left in our collection. Additional sampling of *M. madagascariensis*, especially from the Ranomafana population, and of *M. pulchra* is needed to infer the degree of differentiation of these populations and possibly propose a new taxonomic status.

### 4.4. The Effect of Deforestation on the Demographic History of the Swamp Forest Species

Inferring the demographic history of one species is crucial to better understand how present-day genetic diversity is distributed [84]. A recently published study reveals that both climate change and human colonization triggered habitat loss and fragmentation in Northern Madagascar, two factors that affected the demography of two lemur species [85]. Although it is still widely debated, recent estimates find support for first human activities in Madagascar having started only about 1350 B.P. [43]. Similarly, recent palynological evidence suggests that large scale deforestation in Madagascar occurred later than argued previously, with rises in charcoal abundance older that 2000 y. B.P. being available only from the arid southwest, where natural firing is expected; whereas in wetter regions (such as along the eastern rainforest belt) steep rises in charcoal abundance are mostly found from around 1150–950 y B.P. [86,87]. Similarly interesting, it seems that almost three-quarters of the remaining primary forest of Madagascar was cleared during Colonial Madagascar times (from 1895 to 1925) when population growth was slow and shifting cultivation banned [88], an imposed change in land use that forced many people to find refuge in the forests and oblige them to survive there as shifting cultivators [88]. Finally, Harper et al. [89] reports that Madagacar’s ecosystems have been deeply affected by human-mediated habitat loss and fragmentation in the recent decades, with about the 50% remaining forest cover loss since the 1950s [89,90]. Our demographic analyses identified a severe pattern of demographic contraction affecting the three analyzed swamp forest species (Figure 8) and suggest that this contraction took place in the recent past. Although we cannot exclude that our estimates of effective population size may have been inflated by alleles introduced into the populations by migrants from other populations, it is unlikely that this pattern would have led to a concordant reconstruction of bottlenecks in all populations as found by our analyses. Our probability densities are mostly distributed between 1000 and 100 y ago in almost coincidence with early wide-scale deforestation, and maintained at least until 100 y BP, in accordance with other evidences for increased trends in forest loss and fragmentation in more recent times [88,89], suggesting that forest loss was probably a major trigger for the demographic contraction observed in these species. In addition to this, it is known that these frog species are affected by collecting for the pet trade; about 230,000 specimens of *Mantella* were exported from Madagascar from 1994 to 2003 [29,30], with *M. aurantiaca* being the third worldwide-most traded amphibian species [30]. However, the onset of large-scale collection for the pet trade is more recent than the estimated demographic bottlenecks in the three swamp frog species, which instead are in temporal coincidence with the massive events of forest loss in the past centuries [86,87,88,89]. While also today, habitat loss is certainly the most important and imminent threat to *Mantella* populations (and biodiversity in general), pet trade over-collecting can exacerbate this threat and therefore requires careful regulation. Our data exemplify how the massive past forest reduction has drastically reduced the effective population size of amphibian species, and highlight the importance of fine-scale conservation genetic assessments of the remaining fragmented populations of these species.

## Figures and Tables

**Figure 1 genes-10-00317-f001:**
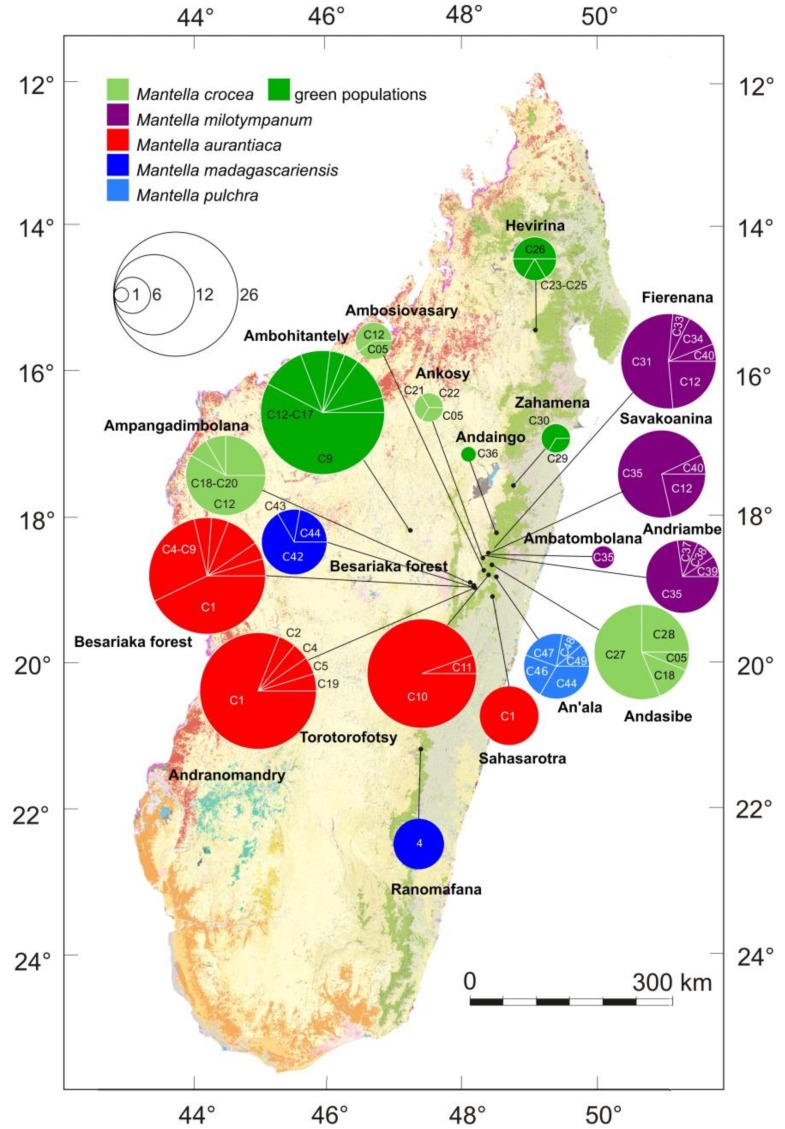
Map with sampling localities and cytochrome b (*COB*) haplotypic frequencies. Original layers from the background map from the Madagascar Vegetation Mapping project: orange, reddish and green colors mark remaining primary vegetation.

**Figure 2 genes-10-00317-f002:**
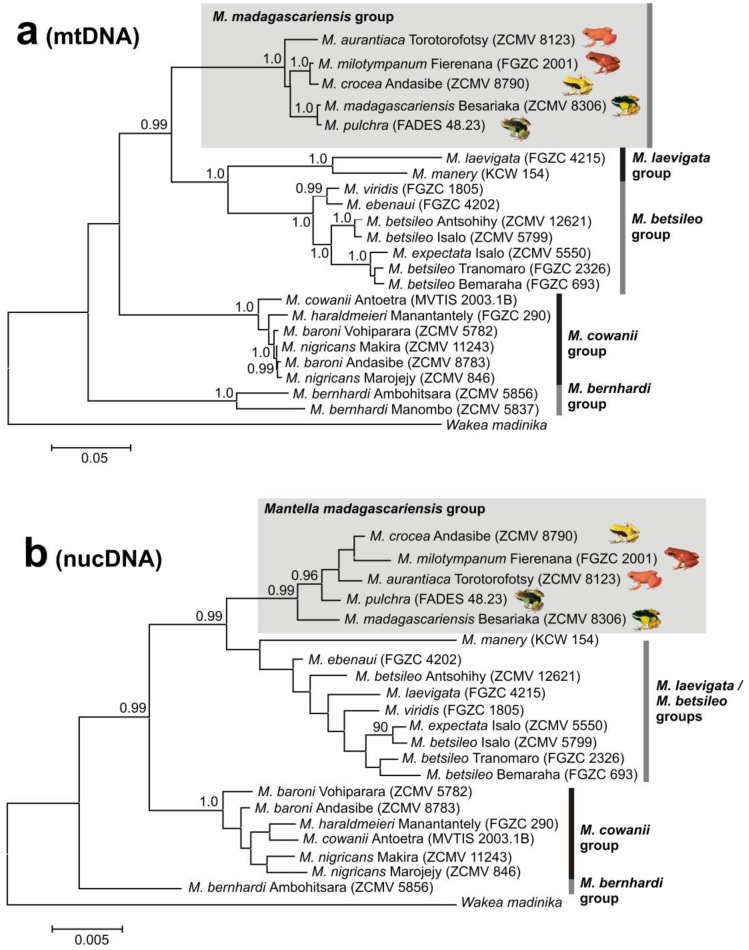
Phylogenetic trees of the genus *Mantella*, showing the position of the target species in the *Mantella madagascariensis* group, obtained by Bayesian Inference analysis of (**a**) 2531 bp of four mitochondrial genes (*COB*, cytochrome oxidase subunit 1 (*COX1*), 12S, 16S), and (**b**) 1710 bp of three nuclear genes (recombination activating gene 1 (*RAG-1*), recombination activating gene 2 (*RAG-2*), brain-derived neurotrophic factor (*BDNF*)). Values at nodes are Bayesian posterior probabilities (not shown if <0.9).

**Figure 3 genes-10-00317-f003:**
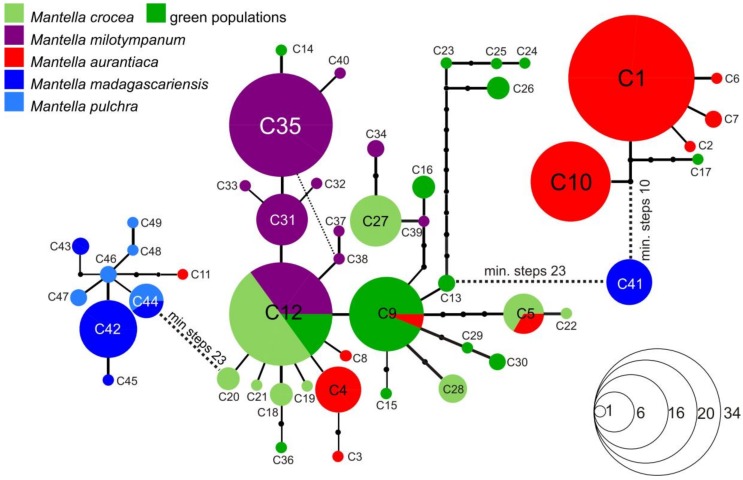
Haplotype network of 49 unique *COB* haplotypes of *Mantella crocea* (green; darker green highlights the haplotypes found in the populations that show the green phenotype), *M. milotympanum* (orange), *M. aurantica* (red), *M. madagascariensis* (blue) and *M. pulchra* (light blue), based on the analysis of a 552-bp DNA fragment of this mitochondrial gene. Size of circles is proportional to the number of individuals sharing a given haplotype (haplotype frequencies as in the inset legend). Shared haplotypes are indicated by circles, with the area being proportional to the number of individuals sharing that haplotype. Inferred intermediate haplotypes are indicated by small black dot.

**Figure 4 genes-10-00317-f004:**
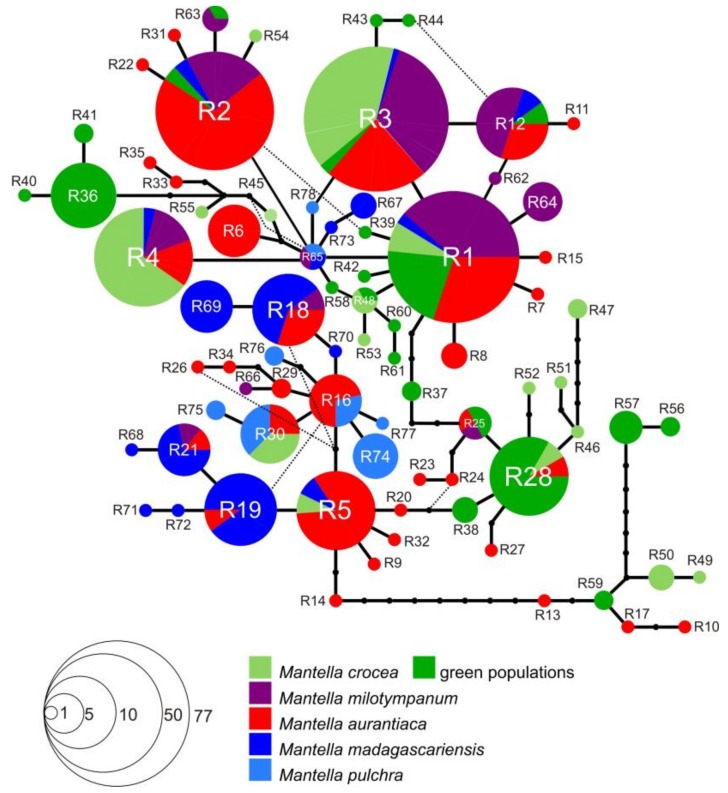
Haplotype network of 78 unique haplotypes of *Mantella crocea*, *M. milotympanum*, *M. aurantica*, *M. madagascariensis* and *M. pulchra* (color code as in Figure 1 and Figure 3), based on the analysis of a 732-bp fragment of the *RAG-1* gene. Size of circles is proportional to the number of individuals sharing a given haplotype. Shared haplotypes are indicated by circles, with the area being proportional to the number of individuals sharing that haplotype (haplotype frequencies as in the inset legend). Inferred intermediate haplotypes are indicated by small black dots.

**Figure 5 genes-10-00317-f005:**
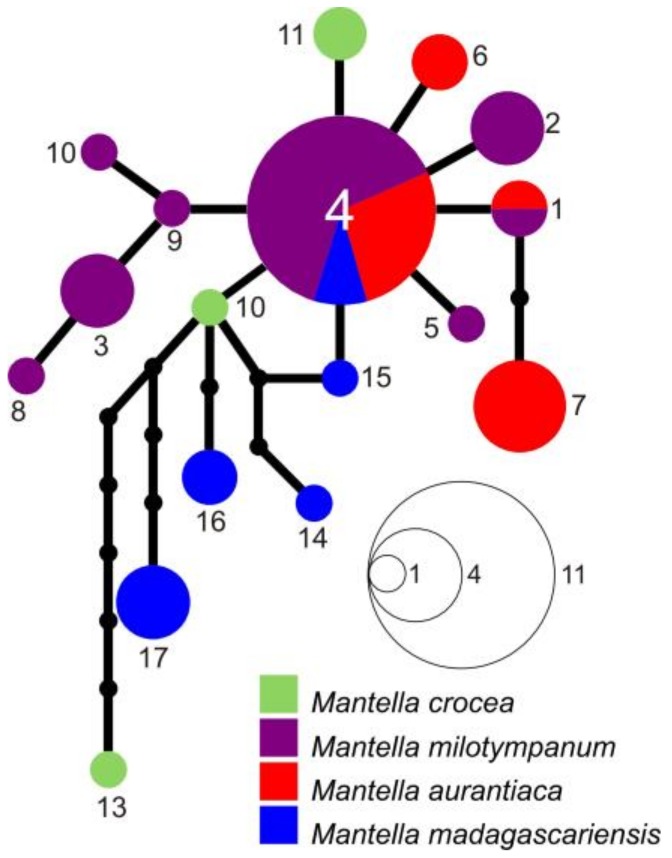
Haplotype network of 17 unique haplotypes of *Mantella crocea*, *M. milotympanum*, *M. aurantica*, *M. madagascariensis*, and *M. pulchra* (color code as in Figure 1, Figure 3 and Figure 4), based on the analysis of a 531-bp fragment of the *RAG-2* gene. Size of circles is proportional to the number of individuals sharing a given haplotype. Shared haplotypes are indicated by circles, with the area being proportional to the number of individuals sharing that haplotype (haplotype frequencies as in the inset legend). Inferred intermediate haplotypes are indicated by small black dots.

**Figure 6 genes-10-00317-f006:**
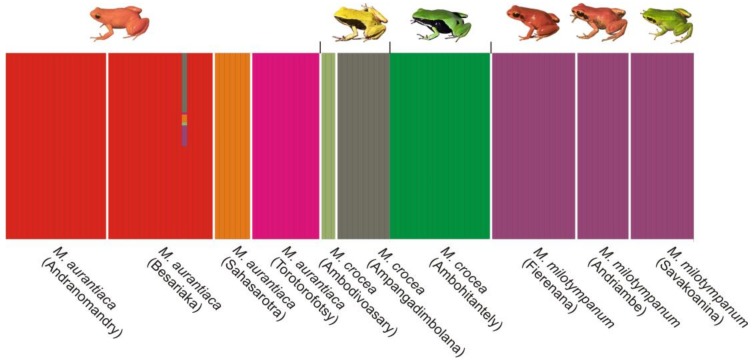
Bayesian analysis of the selected populations of the *M. madagascariensis* group used for the analysis of population structure. Barplot of the clustering solution with the highest posterior probability (K = 7). Each individual in the analysis is represented by a vertical bar, and groups of individuals of the same sampling site are grouped in boxes with an associated label below. The only significantly admixed individual (from the Besariaka population of *M. aurantiaca*) presents its genotype partitioned in colors according to the proportion of it that resembles each of the clusters (e.g., 32% brown reflecting that 32% of the genotype matches that of the *M. crocea* cluster of Ampangadimbolana).

**Figure 7 genes-10-00317-f007:**
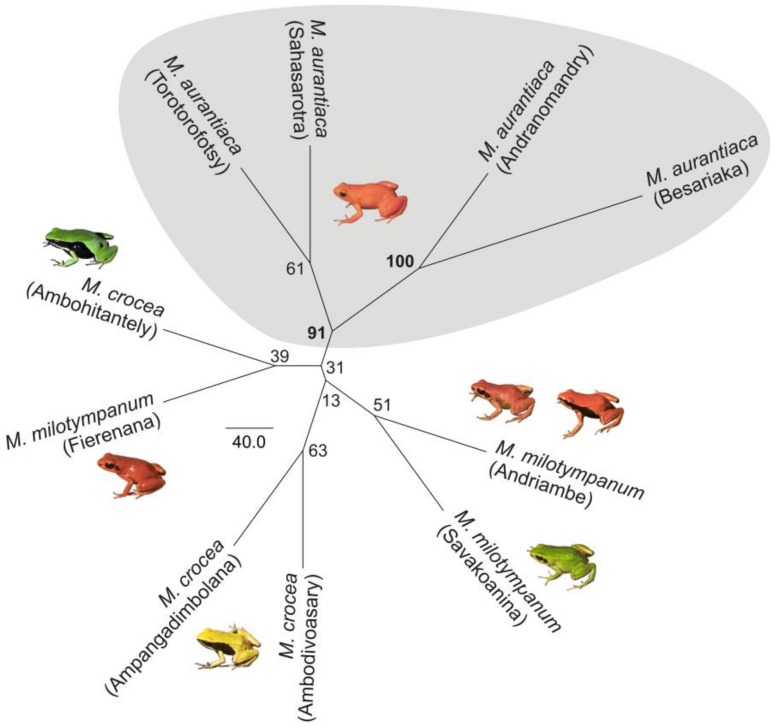
Dendrogram of the sampling localities used for the population structure analyses with nine microsatellite loci. Only the four analyzed populations of *M. aurantiaca* form a substantially supported cluster (bootstrap support 91%).

**Figure 8 genes-10-00317-f008:**
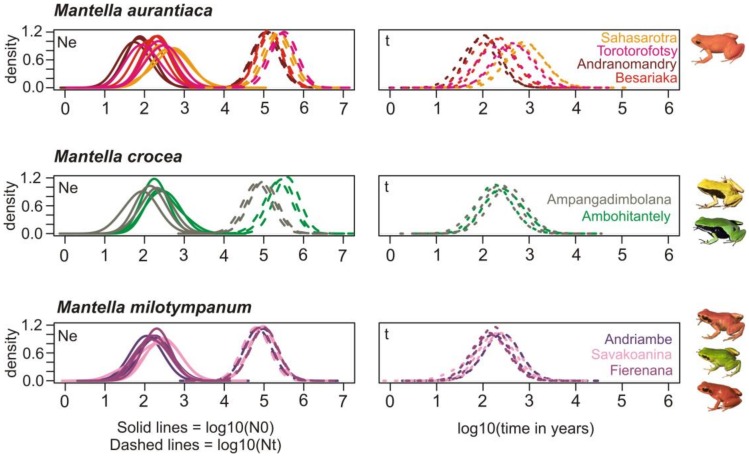
Demographic history of the analyzed *Mantella* spp. populations. For each sampling locality the estimated current effective population size (solid lines; Ne graph), the ancestral effective population size (dashed lines; Ne graph) and the time of the bottleneck (dashed lines; t graph) are shown. Each population presents three curves corresponding to the posterior distribution using three different priors reflecting a bottleneck, expansion and stable demographic scenarios. Each sampling locality is represented with a different color (name color key is shown in the t graphs). Ne, effective population size; t, time in years.

**Table 1 genes-10-00317-t001:** Distribution of sample size per site and species for the population genetic analyses. n: sample size; ANAP: average number of alleles per locus; H_O_: observed heterozygosity; H_E_: expected heterozygosity; F_IS_: inbreeding coefficient.

Taxon	Locality	n	ANAP	H_O_	H_E_	F_IS_
*M. aurantiaca*	Andranomandry	19	10.4	0.7	0.87	0.18
	Besariaka	20	12.78	0.7	0.86	0.18
	Sahasarotra	7	6.11	0.55	0.77	0.25
	Torotorofotsy	13	8.78	0.59	0.79	0.22
*M. crocea*	Ambodivoasary	3	3.44	0.59	0.76	0.12
	Ambohitantely	19	11.89	0.69	0.88	0.21
	Ampangadimbolana	10	6.78	0.46	0.83	0.45
*M. milotympanum*	Fierenana-Sahamarolambo	16	10.67	0.69	0.88	0.21
	Andriambe	10	7.33	0.69	0.85	0.18
	Savakoanina	12	9.56	0.74	0.87	0.14

**Table 2 genes-10-00317-t002:** Pairwise F_ST_ between sampling localities. Fields in white correspond to within species comparisons, and fields in grey to between species comparisons. F_ST_ values can be found above the diagonal, Bonferroni corrected *p*-values are provided below the diagonal with non-significant *p*-values (i.e., *p*-values > 0.05) indicated as n.s. aur—*M. aurantiaca*; croc—*M. crocea*; mil—*M. milotympanum*.

		1	2	3	4	5	6	7	8	9	10
**1**	Aur—Andranomandry	0	0.02	0.05	0.06	0.07	0.03	0.06	0.04	0.05	0.05
**2**	aur—Besariaka	n.s.	0	0.05	0.05	0.08	0.04	0.05	0.05	0.07	0.07
**3**	aur—Sahasarotra	n.s.	0.0045	0	0.06	0.14	0.08	0.08	0.07	0.09	0.10
**4**	aur—Torotorofotsy	0.0045	0.0045	0.0045	0	0.11	0.07	0.07	0.07	0.10	0.10
**5**	croc—Ambodivoasary	n.s.	0.0045	n.s.	n.s.	0	0.06	0.06	0.09	0.09	0.11
**6**	croc—Ambohitantely	0.0045	0.0045	0.0045	0.0045	0.0045	0	0.05	0.04	0.06	0.05
**7**	croc—Ampangadimbolana	0.0045	0.0045	n.s.	0.0045	n.s.	0.0045	0	0.04	0.07	0.05
**8**	mil—Fierenana-Sahamarolambo	0.0045	0.0045	0.0045	0.0045	0.0045	0.0045	n.s.	0	0.05	0.04
**9**	mil—Andriambe	0.0045	0.0045	0.0045	0.0045	n.s.	0.0045	0.0045	0.0045	0	0.04
**10**	mil—Savakoanina	0.0045	0.0045	0.0045	0.0045	n.s.	0.0045	0.0045	0.0045	n.s.	0

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
