# Peer review of "Mitochondrial Introgression, Color Pattern Variation, and Severe Demographic Bottlenecks in Three Species of Malagasy Poison Frogs, Genus Mantella"

_genes, 2019, doi:10.3390/genes10040317_

Round 1
Reviewer 1 Report
p.p1 {margin: 0.0px 0.0px 0.0px 0.0px; font: 12.0px 'Helvetica Neue'} p.p2 {margin: 0.0px 0.0px 0.0px 0.0px; font: 12.0px 'Helvetica Neue'; min-height: 14.0px}Overall this is a thorough and well-constructed work. However, the manuscript is very long, laborious, and complicated. It would be good if another intense round of editing could move some text to a Supplement, and clarify and streamline the main text.
Unclear how and why they change the conception of milotympanum and crocea; more detail on the allocation and delimitation of the populations is needed.
Table 1 is excessively detailed and difficult to interpret.
Table 2 is worse in this regard; move to supplement?
Networks: need to use an ML/MPL method in PhyloNet or SNaQ; SplitsTree is not really a “biological” network. This could seriously impact our interpretation of population divergence and interaction.
It isn’t always clear how and why some methods were chosen and implemented over others; the hypothesis-testing framework isn’t clear. Particularly the demographic methods; why were these bottlenecks assessed this way, and not another method such as skyline plots?
The TCS haplotype networks are very difficult to interpret (they always have been; this isn’t the fault of the authors); is there a way that this information could be better summarized in a phylogeny figure?
Author Response
Response to Reviewer #1 Comments
[Reviewer#1]: Overall this is a thorough and well-constructed work. However, the manuscript is very long, laborious, and complicated. It would be good if another intense round of editing could move some text to a Supplement, and clarify and streamline the main text.
[Authors]: Although we tend to disagree with this evaluation we accept the suggestions and have deleted some parts, and moved some others that were previously included in the main text to the SM section. E.g.: lines 71-81 were entirely removed; Table 1 and Table 2 were moved to SM.
[Reviewer#1]: Unclear how and why they change the conception of milotympanum and crocea; more detail on the allocation and delimitation of the populations is needed.
[Authors]: The conception of milotympanum and crocea follows their currently accepted definition (a summary of which is available in Glaw & Vences 2007) but in order to make this even more clear we applied some changes to the text where we provide the rationale for the allocation of all analysed populations. Previous lines 90-173 (please note that for unknown reasons the line numbering contain a gap between lines 99 and 170) now read as follows: Mantella aurantiaca is characterized by a uniform and translucent orange colouration (Table S1). Mantella milotympanum from the type locality near Fierenana is opaque red orange with a black tympanic region (the characteristic that is also giving the name to this species) and a ventral surface uniformly orange. Mantella crocea shows a lateral black colouration, a frenal stripe, a ventral black colour with a variable number of markings that can be grey, bluish-white or yellowish, and a dorsal colour being yellow-brown in the more southern populations and bright green in the populations of the western and northern portion of its range. Populations with intermediate patterns between Mantella crocea and M. milotympanum (Ambatombolana, Andriambe, Savakoanina and Andaingo [35]) were assigned to M. milotympanum (Ambatombolana, Andriambe, Savakoanina) based on the presence of the black tympanic region and a uniformly orange ventral surface, and based on the lack of both lateral black colouration and frenal stripe. The population of Andaingo was assigned to M. crocea based on the presence of a lateral black colouration, a frenal stripe and a ventral black colouration with sparse grey, yellowish-white markings (see Table S1 for all details).
[Reviewer#1]: Table 1 is excessively detailed and difficult to interpret.
[Authors]: Table 1 was moved to the Supplementary material section
[Reviewer#1]: Table 2 is worse in this regard; move to supplement?
[Authors]: Table 2 was moved to the Supplementary material section
[Reviewer#1]: Networks: need to use an ML/MPL method in PhyloNet or SNaQ; SplitsTree is not really a “biological” network. This could seriously impact our interpretation of population divergence and interaction.
[Authors]: The haplotype networks presented were generated using the software TCS that implements Templeton, Crandall and Sing’s Statistical Parsimony approach (1992, A cladistic analysis of phenotypic associations with haplotypes inferred from restriction endonuclease mapping and DNA sequence data. III. Cladogram estimation. Genetics 132:619-633). Although, we did use splitstree, our goal with this second analysis was to test for the presence of recombination using the Bruen et al.’ Phi test, a methodology that can discriminate effectively between the presence and absence of recombination. With the haplotype networks we want to point the attention to instances of haplotype sharing and haplotype differences which should give a good idea of differentiation between populations. We added the following sentence to clearly specify this. Lines 316-379 (also here please note that for unknown reasons the line numbering contain a gap between lines 319 and 373): “Prior to calculating haplotype relationships using networks, we analysed the nuclear RAG-1 and RAG-2 datasets with the Phi test of Bruen et al. [55] as implemented in SplitsTree4 [56] to asses the presence of recombination in the data. Neither of the genes presented significant evidence of recombination (p-values > 0.1) and thus they were used for network reconstruction. We used TCS version 1.21 [57] to analyse the relationships of the COB haplotypes and of the RAG-1 and RAG-2 inferred haplotypes (Figures 3-5). With these networks we show instances of haplotype sharing and highlight haplotype differences between and within analysed populations, providing an intuitive graphic representation of the observed differentiation between populations, but being aware that phased haplotypes might not in all cases be correct due to the probabilistic nature of the Phase algorithm, and due to the large haplotype variation, the reconstructed haplotype relationships might not be fully accurate.”
[Reviewer#1]: It isn’t always clear how and why some methods were chosen and implemented over others; the hypothesis-testing framework isn’t clear. Particularly the demographic methods; why were these bottlenecks assessed this way, and not another method such as skyline plots?
[Authors]: We acknowledge the existence of multiple methods, but we feel that sometimes as long as the used method is reliable, repeatable and is providing statistically robust results, the use of one method over others remains subjective. We used a set of widely accepted and reliable methods to perform our analyses, and in our opinion the selected tools were the best to test our hypothesis (1. test the taxonomic status of M. crocea and M. milotympanum; 2. test if human-mediated deforestation affected the demographic history of these species) given the available data. Regarding specifically the demographic analyses, the reviewer is right that we could have used extended Bayesian Skyline Plots, but we considered this analysis as an alternative, rather than a complement to our analysis, and after disccussion among all co-authors we preferred the use of Msvar over the extended Bayesian Skyline Plot which will not add additional relevant information for the main questions we want to address.
[Reviewer#1]: The TCS haplotype networks are very difficult to interpret (they always have been; this isn’t the fault of the authors); is there a way that this information could be better summarized in a phylogeny figure?
[Authors]: We agree with the reviewer that haplotype networks are not always clear, especially when lots of haplotypes are present. The requested phylogenetic reconstruction is available as Figure S1 and it parallels the information of the COB haplotype network. However, as we are interested in showing the extensive haplotype sharing between the individuals analysed across the five species (as now more clearly stated in the Methods, as justification for the use of haplotype networks) we are fully convinced that the current RAG-1 and RAG-2 haplotype networks better depict these relationships than a phylogenetic tree, providing further support to our conclusions regarding the presence of incomplete lineage sorting in the RAG-1 and RAG-2 data for these taxa.
Reviewer 2 Report
An interesting complex, which will mean the correct answer may not be simple. I feel the authors need to improve their approach to the analysis. Thereafter the language needs some attention. For now just a few difficulties that should be addressed.
From the data, it is clear that these frogs have a fair amount of isolation between them, but some-times there is some leakage between “species”. This scenario means the group is complex and it is incorrect to assume that throwing more genetic test at it, as suggested in line 728, will give a clearer picture and delineation between species. This situation also makes it problematic to apply a coalescent to estimate population sizes. The reason is that the assumption is that any two lineages can coalesce in the previous generation, but in this situation, genes in species one cannot coalesce with genes in species two, most of the time. As a result, if one look at one species’ variation and reconstruct the past, but now and then another gene pop into existence from another population, then one will tend to over-estimate the size of the population, but by how much? The current inference is logical but if it did not agree with our knowledge of the past, we may be tempted to dismiss it because of this breach of assumption. So, while it’s plausible, I do not find this very convincing.
Pairwise Fst’s: Line 575, both being lower seems incorrect? Comparison in 576. Pairwise Fst’s are not independent of each other, so if A to B is small, and if A to C is large, then B to C is also likely to be large, for this reason matrixes of Fst cannot be compared as if they are all independent, as is done with a t-test.
Haplotype inference for nuclear genes: In the case of RAG 1 it involves grouping 14 polymorphic sites into haplotypes. Seeing as there is 2^14 = 16384 possible haplotypes, it is important to explain this step clearly. The current info is not sufficient to be repeatable. And I am nervous of what one can confidently conclude from it given potential leakage over species lines and the introduction of very different haplotypes. It is unclear to me how important breaking of assumptions would be on the conclusions.
The fact that figure 6 group each frog as a separate entity makes a good argument that they are for the most isolated. However, looking for admixture, K should be reduced until before each is a separate entity, and then admixture will be clearer.
Author Response
Response to Reviewer #2 Comments
[Reviewer#2]: An interesting complex, which will mean the correct answer may not be simple. I feel the authors need to improve their approach to the analysis. Thereafter the language needs some attention. For now just a few difficulties that should be addressed. From the data, it is clear that these frogs have a fair amount of isolation between them, but some-times there is some leakage between “species”. This scenario means the group is complex and it is incorrect to assume that throwing more genetic test at it, as suggested in line 728, will give a clearer picture and delineation between species.
[Authors]: The reviewer here is right and we decided to modify this sentence as follows (lines 923-925): “… the current data do not allow to conclusively delimit independent evolutionary units (e.g. the taxonomic status of M. crocea and M. milotympanum populations) and we do not see a simple answer in immediate reach.” (We removed the part that suggests that adding more genetic test will provide a clearer picture).
[Reviewer#2]: This situation also makes it problematic to apply a coalescent to estimate population sizes. The reason is that the assumption is that any two lineages can coalesce in the previous generation, but in this situation, genes in species one cannot coalesce with genes in species two, most of the time. As a result, if one look at one species’ variation and reconstruct the past, but now and then another gene pop into existence from another population, then one will tend to over-estimate the size of the population, but by how much? The current inference is logical but if it did not agree with our knowledge of the past, we may be tempted to dismiss it because of this breach of assumption. So, while it’s plausible, I do not find this very convincing.
[Authors]: We thank the reviewer for highlighting this problem. We agree that migration events between localities will increase the local genetic diversity by the introduction of new alleles, and thus could distort the results toward presenting a larger effective population size that there is in reality. Such pattern could be observed as an inferred population expansion with the current effective population size being larger than the ancestral due to the migration events, and consistently the time of the expansion may be reflective of the time of migration (rather than an actual demographic expansion). Approaches such as MsVar are likely to fail (stall) at computing genealogies inclusive of alleles deriving from migration events as i) the time of coalescent of the immigrant allele requires additional time for the allele to move from the population where it was found to its original population, and ii) then it needs to wait until the two population coalesce; therefore the coalescence time from such an allele would be much deeper in time than the coalescences of the locality causing problems to the software. We did not observe such problems in our runs. Lastly, for migrants to have an effect on the local gene genealogies the number of migrations from locality B into A should probably be large, as otherwise the newly introduced alleles may disappear via genetic drift.
The results of our analyses rather than representing the outcome of a population expansion (as it would be expected with alleles migrating among localities) are indicative of a bottleneck that can be observed in all localities analysed. While the absolute numbers inferred for the effective population size may be slightly larger than in reality due to any migration that may have occurred between localities, the overall trend in effective population size across time should be indicative of the actual effective population size.
We have added a sentence on line 778 and in lines 1069-1072 stating this potential problem and our interpretation according to which it is unlikely to have a strong effect on our main conclusions.
[Reviewer#2]: Pairwise Fst’s: Line 575, both being lower seems incorrect? Comparison in 576. Pairwise Fst’s are not independent of each other, so if A to B is small, and if A to C is large, then B to C is also likely to be large, for this reason matrixes of Fst cannot be compared as if they are all independent, as is done with a t-test.
[Authors]: We thank the reviewer for highlighting this error. As the number of between species comparisons is much larger than the within species comparisons we agree that we could not implement a statistical test that enables the use of non-independent data. Thus we rephrased our statement in Line 746-748 as follows: “The FST values were overall low both within and between species, with the average within species FST (0.05) slightly smaller than the average between species FST (0.07).”
[Reviewer#2]: Haplotype inference for nuclear genes: In the case of RAG 1 it involves grouping 14 polymorphic sites into haplotypes. Seeing as there is 2^14 = 16384 possible haplotypes, it is important to explain this step clearly. The current info is not sufficient to be repeatable. And I am nervous of what one can confidently conclude from it given potential leakage over species lines and the introduction of very different haplotypes. It is unclear to me how important breaking of assumptions would be on the conclusions.
[Authors]: We thank the reviewer for pointing this out. We provide additional information on the performed analysis to improve the reproducibility of our computations. Lines 285-291 where modified as follows: “Because the RAG-1 and RAG-2 alignments contained heterozygous positions we phased genotypes to identify haplotypes for further analyses using the PHASE algorithm (version 2.1.1) with default settings [47] as implemented in the software DnaSP (version 6.12.01; [48]). PHASE parameters were 1000 iterations, one thinning interval and 100 burn-in iterations and a posterior probability threshold of 0.9 to determine the most probable inferred haplotypes for each nuclear sequence. Analyses were repeated three times using different seed values.”
We also reiterate (as is now more clearly stated in the manuscript; Mat&Met on haplotype network reconstruction) that the purpose of the haplotype networks, and of the nuclear gene analyses overall, is to visualize the widespread sharing of haplotypes across the putative species in the complex (lines 374-379).
[Reviewer#2]: The fact that figure 6 group each frog as a separate entity makes a good argument that they are for the most isolated. However, looking for admixture, K should be reduced until before each is a separate entity, and then admixture will be clearer.
[Authors]: Figure 6 shows that the population of M. aurantiaca from Andranoandry and Besariaka are likely forming a unit, and that the analysed populations of M. milotympanum altogether also form a unit, while the other two populations of M. aurantiaca and the three populations of M. crocea are separate entities. We have now added a figure (Figure S2) to the supplementary material showing the results of progressive portioning of the data for values of K=2 until K=7, and we report on these analyses in the Results (lines 698-710; please note that for unknown reasons the line numbering contain a gap between lines 698 and 704). With Figure S2 we show that the populations form clear clusters without admixture across the different values shown and only single individuals are showing admixture. Furthermore, the identity of admixed individuals and their admixture proportions and cluster assignment change across different K values tested, suggesting that these observations are spurious results.
Round 2
Reviewer 1 Report
This revision and the responses are substantially improved, and I recommend publication after minor revisions. It would be nice if the authors included more sophisticated analyses of gene flow, such as estimating the magnitude and timing of migration and hybridization (migrate, treemix, snaq, etc.) but this is not crucial. Great to see such an in-depth phylogeographic estimate from this group and region!
Author Response
Response to Reviewer #1 Comments
[Reviewer#1]: This revision and the responses are substantially improved, and I recommend publication after minor revisions. It would be nice if the authors included more sophisticated analyses of gene flow, such as estimating the magnitude and timing of migration and hybridization (migrate, treemix, snaq, etc.) but this is not crucial. Great to see such an in-depth phylogeographic estimate from this group and region!
[Authors]: Thank you for this positive evaluation. The tight deadlines make it impossible to add extensive additional statistics; however, we considered the useful suggestions of the reviewer, and performed at least one of the suggested analyses: An estimate of migrants using BayesAss software revealed limited recent migration between the studied populations, thus confirming our results of limited gene flow among them. We have added a few sentences to methods (lines 440-449), results (761-764 ) and discussion (922) reporting this additional analysis.